# Deep Learning for Liver Disease Stratification: Findings from UKBB MRI

**Rami Al-Belmpeisi**[1,2] iD                                                    RALBE@DTU.DK
**Josefine Vilsbøll Sundgaard**[1,2] iD
**Peter Hjørringgaard Larsen**[2]
**Anders Bjorholm Dahl**[1] iD

[1] *Department of Applied Mathematics and Computer Science, Technical University of Denmark, Denmark*

[2] *Novo Nordisk, Copenhagen, Denmark*

**Editors:** Accepted for publication at MIDL 2026

## Abstract

Metabolic dysfunction-associated steatotic liver disease (MASLD) and its progressive form, metabolic dysfunction-associated steatohepatitis (MASH), have become more prevalent, spurring interest in using magnetic resonance imaging (MRI) sequences for diagnosis. In this study, we propose a method that uses deep learning to diagnose MASLD and MASH with significant fibrosis from single-slice (2D) and volumetric (3D) MRI sequences that originate from the UK Biobank. In this paper, we focus on transparent decision-making. Our study shows that imposing anatomically informed constraints by using a liver segmentation mask on the network's input has minimal impact on diagnostic performance. Still, it redirects attention to clinically relevant liver regions, preventing shortcut learning from extrahepatic features, such as subcutaneous fat. These constraints shift the focus of the model toward proton density fat fraction (PDFF) maps for healthy liver assessment, $T_1$ maps for MASLD diagnosis, and both sequences to identify MASH with significant fibrosis. Our top-performing models achieve AUCs of 0.89/0.96/0.79 for the diagnosis of the healthy/MASLD/MASH groups with significant fibrosis, respectively. Despite label noise and limited sequence specificity, which primarily hinder predictive performance in cases of MASH with significant fibrosis, the identified indicators are frequently located in liver regions consistent with prior understanding of disease progression. In conclusion, we find that 2D MRI sequences are sufficient for diagnosing MASLD/MASH with significant fibrosis, as performance decreases and computation time increases when using 3D volumes.

**Keywords:**
MASLD, MASH, Magnetic Resonance Imaging, Imaging biomarkers

## 1. Introduction

A growing percentage of people suffer from metabolic dysfunction-associated steatotic liver disease (MASLD). The prevalence of MASLD has increased drastically over the past thirty years, from 17.6% in 1990 to 23.4% in 2019 (Paik et al., 2023). Moreover, the prevalence of MASLD among adults is currently estimated to be 38% (Younossi et al., 2024). This introduces a burden on the health system, considering that MASLD is the root cause of both chronic liver disease and liver-related complications (Chan et al., 2023). Fibrosis in MASLD and MASH is typically assessed using the Kleiner/Brunt scoring system (Kleiner et al., 2005), which is based on 14 histological features and ranges from no fibrosis (F0),

minimal (F1), periportal (F2), bridging (F3), to cirrhosis (F4). MASLD is defined as a liver fat excess ($\geq 5.5\%$) accompanied by at least one of five cardiometabolic risk factors and is characterised by little to no fibrosis (F0-F1). The five cardiometabolic risk factors include obesity, dysglycemia or hyperglycemia, arterial hypertension, hypertriglyceridemia, and low levels of high-density lipoprotein cholesterol, according to recent consensus (Rinella et al., 2023). Its progressive form, metabolic dysfunction-associated steatohepatitis (MASH), is accompanied by liver inflammation, cell injury, and progressive fibrosis ($\geq$F1) (Dulai et al., 2017). The rate of progression from MASLD to MASH with significant fibrosis is slow, as fibrosis typically develops by 0.07 and 0.14 stages annually for MASLD and MASH, equivalent to one stage of fibrosis progression in over 14 and 7 years, respectively (Singh et al., 2015).

An early and accurate diagnosis is critical to the success of treatment for MASLD and MASH, while presently, the only established means for diagnosing MASLD and MASH is liver biopsy. However, it is an expensive and invasive procedure that involves risks of bleeding and infections and can lead to misdiagnosis due to sampling errors. The shortage of broadly established non-invasive diagnostics for MASLD and MASH has redirected attention to image-derived biomarkers, particularly from MRI sequences (Cathcart et al., 2025).

Several liver conditions can be detected using different MRI sequences, with varying accuracy. A liver with MASLD is characterised by excess liver fat, which can be quantified by Proton Density Fat Fraction (PDFF) maps (Azizi et al., 2025; Kinner et al., 2016). The MASH liver becomes stiff due to the onset of fibrosis, and liver stiffness can be measured using Magnetic Resonance Elastography (MRE) (Berzigotti et al., 2021; Dulai et al., 2016; Singh et al., 2016). A liver with MASH exhibits fibrosis, scarring and fat infiltration, pathological conditions that can be identified in native spin-lattice relaxation maps ($T_1$) (Nauffal et al., 2024). The iron-corrected $T_1$ ($cT_1$) sequence (Dennis et al., 2020) further improves diagnostic specificity compared to $T_1$, as it accounts for the hepatic iron content.

Building on these individual sequences, recent research has explored multi-modal approaches to improve diagnostic accuracy (Kim et al., 2020; Dennis et al., 2021; Al-Belmpeisi et al., 2024). While the sequences used and their multi-modal combinations are still clinically relevant, the recent transition of the nomenclature from Non-Alcoholic Fatty Liver Disease (NAFLD) and Non-Alcoholic Steatohepatitis (NASH) to MASLD/MASH (Rinella et al., 2023) underscores the necessity of keeping diagnostic frameworks up-to-date with the medical community for their applicability. A thorough comparison among MRI sequences is significantly hindered by different diagnostic assumptions and variations in data availability across studies (Cathcart et al., 2025).

We compare using the UKBB dataset, which provides a unique combination of a large-scale cohort with MRI availability, aiming towards transparent, automated diagnosis of MASLD and MASH with significant fibrosis. Comparisons across studies are frequently challenging because they rely on different criteria for identifying MASLD and MASH, given data availability. This challenge is also present in UKBB, as liver histology, the "imperfect" gold standard of diagnosis, is not available. While the International Classification of Diseases (ICD9/10) codes are often used instead, the significant discrepancy between disease incidence and global prevalence estimates in UKBB indicates substantial underdiagnosis (Hayward et al., 2021). Therefore, we utilise serum markers for assessing disease stratification labels for MASLD and MASH with significant fibrosis ($\geq$F2). To mitigate the limited

specificity of serum-based markers in differentiating among liver diseases (Bell et al., 2010; Di Mauro et al., 2021; Dawod and Brown, 2024), ICD-9/10 codes are used to exclude other hepatic pathologies and conditions. Furthermore, the available serum markers are collected during visits preceding the imaging visit, reflecting real-world longitudinal data that are often imperfect. While large temporal gaps between imaging and serum-based disease stratification can introduce label noise, their clinical impact is limited by the slow progression of MASLD and MASH with significant fibrosis, as these diseases typically develop over longer timescales (Singh et al., 2015). Nonetheless, because neural networks tend to overfit on noisy labels (Zhang et al., 2016), this setting offers an opportunity to investigate robustness under realistic label-uncertainty conditions, a recurring and unacknowledged challenge in deploying AI in clinical practice.

This study addresses critical gaps in understanding how different MRI sequences and input configurations impact MASLD/MASH diagnosis in large-scale cohorts. In particular, we compare three input configurations: (1) a full MRI slice with liver coverage, (2) a full MRI slice combined with its segmentation mask, and (3) the liver region isolated MRI slice by masking out extrahepatic context. We also assess the trade-off between 2D and 3D approaches, balancing hepatic heterogeneity against computational cost. Finally, we examine robustness to label noise arising from temporal gaps between imaging and serum-based stratification, a common challenge in real-world, large-scale datasets. Our findings highlight that abdominal context marginally improves diagnostic performance, informing practical strategies for deploying AI-driven liver diagnostics at scale.

## 2. Data

UKBB is a large-scale population study that contains extensive lifestyle, health, and imaging data from about half a million individuals. MRI data under the approved UKBB project number 53,639 are used for MASLD/MASH with significant fibrosis diagnosis, utilising $T_1$/PDFF/fat/water images with respective in-plane resolutions of $1.1/1.7/2.23/2.23$ mm and slice thicknesses of $8/10/3/3$ mm, respectively. The $T_1$ sequence is readily available, and PDFF maps are derived using the Three-Point-Dixon method (Ma, 2008). The framework outlined by Langner et al. (Langner et al., 2020) is used to obtain volumetric neck-to-knee fat, water and fat fraction images, which are based on graph cuts for co-registration as MRI images are scanned in segments (Ekström et al., 2020), resulting in a $811 \times 449 \times 522$ mm volume. Preprocessing steps include manually examining the volumetric images to identify fat–water swaps and exclude affected scans from the analysis, and removing high-frequency background artefacts generated by the three-point Dixon method on the PDFF slices using Otsu thresholding. We also obtain liver masks for the $T_1$, and PDFF 2D slices from Al-Belmpeisi et al. (Al-Belmpeisi et al., 2024).

Our sub-study cohort excludes participants from UKBB with ICD-10 codes for other non-MASLD/MASH-related liver diseases and conditions, such as liver transplant, liver failure, viral hepatitis, etc. We also exclude participants without MRI, blood biomarkers, data to obtain CRFs, and liver PDFF assessment. This results in a study cohort of 18,073 participants, of which 11,706 (64.8%) are healthy, 4,254 (23.5%) have MASLD, and 2,113 (11.7%) have MASH with significant fibrosis. While the MRI sequences originate from the imaging visit (2014-now), the latest available markers originate from the initial assessment

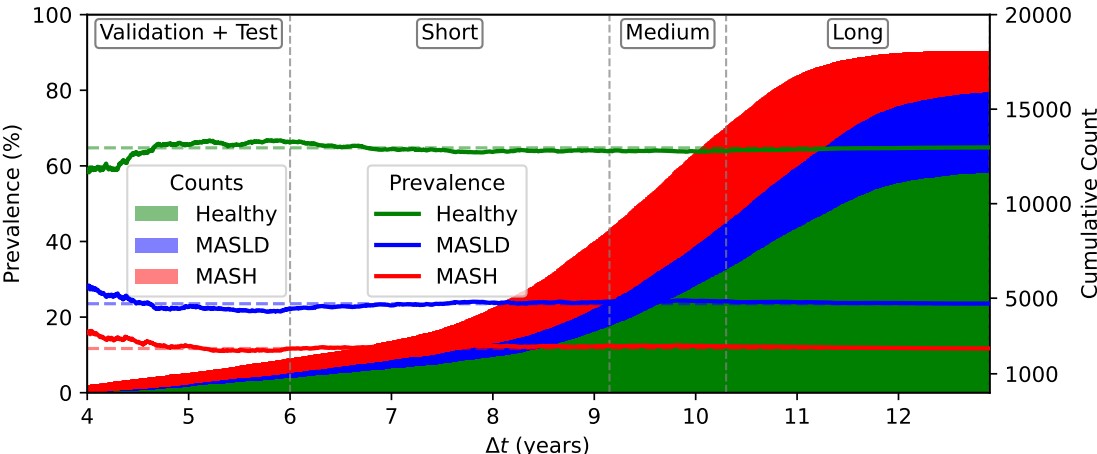

Figure 1: The distribution of temporal misalignment across our cohort. On the left y-axis, the prevalence (%) of each health group (Healthy, MASLD, MASH with significant fibrosis) is displayed for data with time differences between stratification and imaging up to $\Delta t$. On the right y-axis, the cumulative count of cases per class is displayed for increasing $\Delta t$. Vertical bands separate the data into validation and testing data, while the remaining training data is further split into three equally sized temporal segments (Short, Medium, Long).

and first repeat assessment imaging visits (2006-2010, 2012-2013). To mitigate this issue, we construct the validation and test sets from participants with the shortest time between blood collection and imaging, and use the remainder for training, as shown in Figure 1. We define MASLD as liver fat greater than 5.5% in conjunction with at least one CRF. The *cTAG* score is used with a threshold of 0.34, which is reported to meet an area under the receiver operating characteristic curve (AUC) of 0.9 in MASH with significant fibrosis ($\geq$F2) (Dennis et al., 2020).

## 3. Methods

As illustrated in Fig. 2, we explore the potential of deep learning for the automated diagnosis of MASLD and MASH with substantial fibrosis on MRI scans from UKBB. We classify participants into healthy, MASLD, and MASH with significant fibrosis using an 18-layer ResNet architecture (see Appendix B for further details). This model is applied across all input settings—single- and multi-modal (early fusion), and 2D and 3D configurations. All models are trained on a data split of 92%/4%/4% (16579/747/747) for training, testing, and validation, respectively, using the focal loss (Ross and Dollár, 2017) to mitigate class imbalance. Before classification, all image sequences are standardised to zero mean and unit variance. For 2D, we consider single-modal inputs ($T_1$, PDFF) and multi-modal settings ($T_1$+PDFF, fat+water) to enable a direct comparison across models. For 3D, we consider single-modal (fat fraction) and multi-modal (fat+water) inputs.

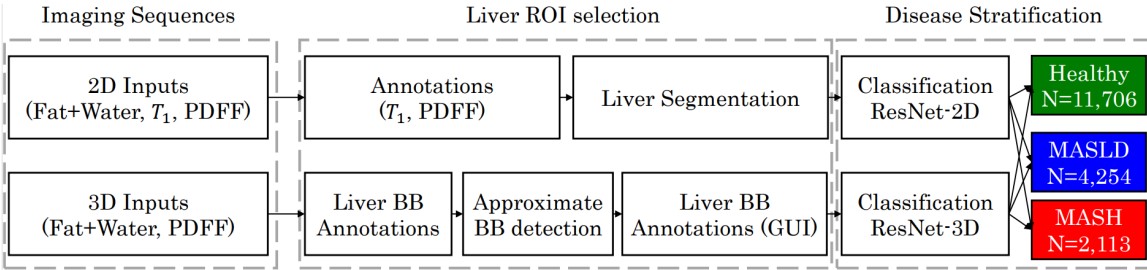

Figure 2: Our proposed diagnostic assessment pipeline. On top, liver masks and subsequent liver segmentation models, described in Al-Belmpeisi et al. (Al-Belmpeisi et al., 2024), are used for the single-slice (2D) approach. The bounding box (BB) annotations are generated using a developed GUI for the volumetric (3D) approach. Using a ResNet architecture, both pipelines classify participants into healthy, MASLD, or MASH.

Before 3D classification, we localise the liver with a 3D bounding box detector and crop the volume around the detected box, reducing the input volume size by $\approx 67\%$ (see Appendix A for further details).

The training data volume size is significantly larger than that for slices. This suggests that 3D-specific data augmentations should be tested to enable a fair comparison between 2D and 3D inputs. Therefore, we use several 3D augmentations at training time, including random translations (up to $\pm 9$ mm) and random rotations (up to $5°$), both constrained to the axial slice plane due to large voxel anisotropy; intensity shifts up to $\pm 0.1$; additive Gaussian noise (mean 0, standard deviation up to 0.1); and gamma-based contrast adjustment with $\gamma$ in the range 0.45–0.50. Each transformation is applied independently with a 10% probability, resulting in augmentations for approximately half of the 3D inputs during training. These augmentations also enhance the generalisability of the trained models by enabling them to learn more robust features through simulating variations in patient positioning, acquisition conditions, and scanner calibration.

Test set performance is reported as the mean $\pm$ standard deviation across four runs per experiment, using different random seeds to control stochastic components such as weight initialisation, the per-epoch shuffle order of training samples, and the stochasticity of data augmentation. A direct comparison of models in the multi-class setting is based on aggregated metrics across health groups, obtained by weighting per-class metrics by the corresponding group sizes.

### 3.1. Decision explainability and role of liver features

While deep learning achieves high diagnostic accuracy, it is essential to verify that decisions rely on biologically relevant regions rather than on spurious correlations, also known as shortcut learning (Geirhos et al., 2020). To accomplish this, we investigate the impact of the different masking strategies employed on decision-attribution regions when applied to 2D inputs ($T_1$, PDFF):

1. **Original Image (OI)**: The network input is the entire MRI image without any information on the liver region. This evaluates whether the model uses global context for disease classification.

2. **Augmented Mask (AM)**: The network input is the entire MRI slice with an additional image channel of the liver segmentation mask. This assesses whether explicit liver localisation improves predictive power by concentrating activations inside the liver.

3. **Liver-Region Isolated (LRI)**: The network input is the MRI slice with only the liver region; the intensity of non-liver regions is set to zero. This forces the model to focus solely on liver features, thereby assessing the dependence of non-liver features on the decision-making.

We display 1% of the most activated image regions, computed as the gradient norm of the predicted class with respect to the input image, using Guided Backpropagation (Springenberg et al., 2014). This specific threshold is selected after manual inspection, as it usually corresponds to a distinct and interpretable area for analysis, and is given by:

$$S_{\text{top 1\%}} = \left\{ \left| \frac{\partial Y_{\text{max}}}{\partial X} \right|_{\text{top 1\%}} \right\}, \tag{1}$$

where $Y_{\text{max}}$ is the predicted class and $X$ refers to the input images. Combining them with decision-making activations, we quantify interpretability using three metrics: (1) fraction of activations within the liver ($P$), (2) mean distance from the liver ($d$), and (3) modality contribution to the final decision ($Q$) in the multi-modal setting.

## 3.2. Label noise experiments

The temporal discrepancy between imaging and serum biomarker collection may raise concerns for our study. It is crucial to quantify the effect of temporal gaps, as they introduce asymmetric noise into serum-based disease-stratification labels. To address this, we split the training dataset into three equal-sized subsets by time gap: 6-9.1, 9.1-10.3, and 10.3-12.9 years. We then train models on all three subsets and examine substantial variation in classification metrics across subsets for each health group. This experiment evaluates whether increasing temporal gaps degrade classification performance and whether models remain robust under realistic longitudinal conditions.

## 4. Results

Table 1 shows the performance of all proposed classification experiments across input modalities and dimensionalities (Dim.), masking techniques, and health groups. A 3D bounding box (BB) is used to crop the region of interest in volumetric inputs, as displayed in the lowermost two columns, which is discussed in greater detail within Appendix A. Notably, our top-performing model ($T_1$+PDFF on OI) achieves per-class AUCs of 0.89/0.96/0.79 for the healthy, MASLD, and MASH with significant fibrosis, respectively, on the test set.

To provide a comprehensive evaluation of the top-performing model, which captures the synergy between $T_1$ and PDFF across health groups, Table 2 presents additional disease-stratification findings.

The decision attribution results on the test set are reported quantitatively in Table 3, with a qualitative visualisation supplied in Fig. 3. Results are expressed in terms of

Table 1: Class-wise AUC performance across input modalities and masking techniques.

| Data | Dim. | Masking | Healthy | MASLD | MASH | Weighted |
|------|------|---------|---------|-------|------|----------|
| $T_1$ | 2D | OI | $0.82 \pm 0.00$ | $0.79 \pm 0.01$ | $\mathbf{0.79 \pm 0.00}$ | $0.81 \pm 0.00$ |
| $T_1$ | 2D | AM | $0.82 \pm 0.00$ | $0.78 \pm 0.01$ | $\mathbf{0.79 \pm 0.01}$ | $0.81 \pm 0.00$ |
| $T_1$ | 2D | LRI | $0.81 \pm 0.01$ | $0.73 \pm 0.02$ | $\mathbf{0.79 \pm 0.01}$ | $0.79 \pm 0.01$ |
| PDFF | 2D | OI | $0.85 \pm 0.00$ | $0.95 \pm 0.01$ | $0.72 \pm 0.01$ | $0.86 \pm 0.00$ |
| PDFF | 2D | AM | $0.85 \pm 0.01$ | $0.95 \pm 0.00$ | $0.73 \pm 0.00$ | $0.86 \pm 0.00$ |
| PDFF | 2D | LRI | $0.86 \pm 0.00$ | $0.94 \pm 0.01$ | $0.72 \pm 0.01$ | $0.86 \pm 0.00$ |
| $T_1$ + PDFF | 2D | OI | $\mathbf{0.89 \pm 0.00}$ | $\mathbf{0.96 \pm 0.01}$ | $\mathbf{0.79 \pm 0.01}$ | $\mathbf{0.89 \pm 0.00}$ |
| $T_1$ + PDFF | 2D | AM | $\mathbf{0.89 \pm 0.00}$ | $0.94 \pm 0.02$ | $\mathbf{0.79 \pm 0.01}$ | $\mathbf{0.89 \pm 0.00}$ |
| $T_1$ + PDFF | 2D | LRI | $0.88 \pm 0.01$ | $0.95 \pm 0.01$ | $0.77 \pm 0.02$ | $0.88 \pm 0.01$ |
| Fat + Water | 2D | OI | $0.86 \pm 0.01$ | $0.90 \pm 0.01$ | $0.77 \pm 0.01$ | $0.86 \pm 0.01$ |
| Fat + Water | 3D | BB | $0.84 \pm 0.01$ | $0.88 \pm 0.01$ | $0.72 \pm 0.01$ | $0.83 \pm 0.01$ |
| PDFF | 3D | BB | $0.85 \pm 0.01$ | $0.89 \pm 0.01$ | $0.73 \pm 0.01$ | $0.84 \pm 0.01$ |

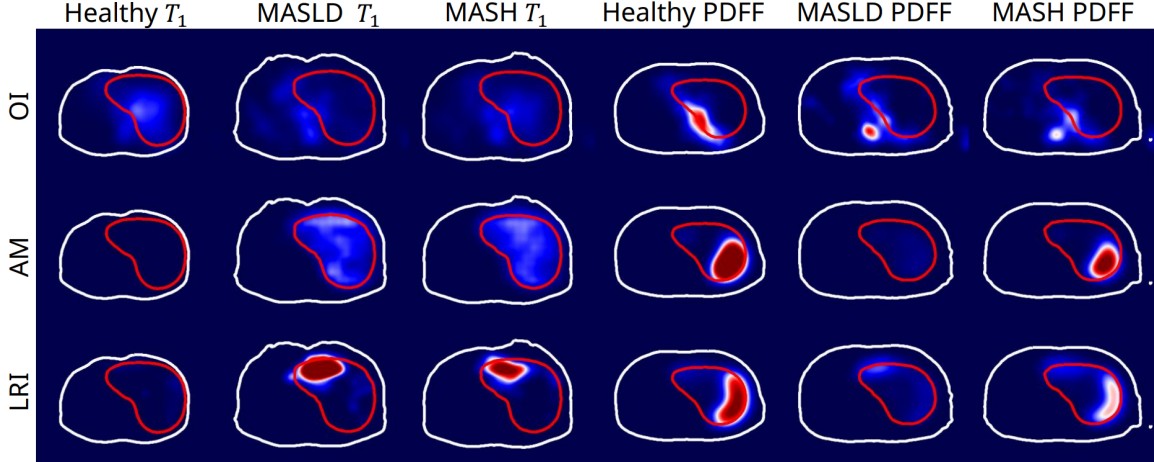

Figure 3: Average saliency maps across the test set for classification networks with three different masking techniques (OI, AM, LRI). $T_1$ sequences are displayed on the three leftmost columns, while PDFF sequences are on the rightmost three. The results are arranged in columns by ground-truth disease class and in rows by masking technique. The average locations of the liver and abdomen across the test set are shown by the red and white contours, respectively.

Table 2: The harmonic mean of precision and recall, denoted as $f_1$, across different sequences on the OI masking technique.

| Data | Healthy | MASLD | MASH | Weighted |
|------|---------|-------|------|----------|
| $T_1$ | $0.84 \pm 0.03$ | $0.20 \pm 0.08$ | $\mathbf{0.50 \pm 0.02}$ | $0.64 \pm 0.02$ |
| PDFF | $\mathbf{0.90 \pm 0.00}$ | $0.59 \pm 0.07$ | $0.34 \pm 0.04$ | $0.75 \pm 0.02$ |
| $T_1 + \text{PDFF}$ | $\mathbf{0.90 \pm 0.00}$ | $\mathbf{0.61 \pm 0.02}$ | $0.42 \pm 0.03$ | $\mathbf{0.77 \pm 0.01}$ |

quantities $P$, $d$ and $Q$, which were defined earlier in Section 3.1, to characterise the prominent regions in connection with the liver.

Last, the label noise experiments on the impact of large time gaps are summarised in Table 4. There, it can be seen that for the healthy group, the AUC is identical across all subsets (0.88-0.89; $H = 0.0$, $p = 1.0$). MASLD showed minor, non-significant variation (0.95-0.96; $H = 2.44$, $p = 0.295$) of performance, while MASH with significant fibrosis exhibits a minor variation (0.95-0.96; $H = 4.79$, $p = 0.091$), which is not large enough to be statistically significant.

## 5. Discussion and Conclusion

### Performance of MRI sequences

The multimodal model on $T_1$+PDFF outperforms other sequences, as revealed by weighted-$f_1$ in Table 2. This is consistent with previous research on UKBB (using the older nomenclature), which identified diagnostic synergy between $T_1$ and PDFF (Al-Belmpeisi et al., 2024). Their claims support our findings, as we also argue that hepatic iron overload counteracts the fibrosis-related signal increase in $T_1$ maps by lowering $T_1$ values, making fibrosis harder to detect. This hampers MASH with a significant fibrosis diagnosis as hepatic iron overload affects 30%-40% of UKBB participants (Mojtahed et al., 2019). Mitigating this issue in the absence of contrast-enhanced MRI would require an independent assessment of hepatic

Table 3: Saliency map metrics for masking techniques.

| Masking | Class | $T_1$ | | | PDFF | | |
|---------|-------|-------|-------|-------|------|-------|-------|
| | | P (%) | d (mm) | Q (%) | P (%) | d (mm) | Q (%) |
| | Healthy | $48.3 \pm 5.8$ | $9.2 \pm 6.0$ | 49.8 | $1.1 \pm 31.4$ | $3.3 \pm 7.8$ | 50.2 |
| OI | MASLD | $34.8 \pm 7.0$ | $26.1 \pm 13.3$ | 48.0 | $78.7 \pm 23.1$ | $20.0 \pm 22.8$ | 52.0 |
| | MASH | $41.3 \pm 9.8$ | $21.2 \pm 15.5$ | 52.5 | $35.0 \pm 43.1$ | $20.3 \pm 22.1$ | 47.6 |
| | Healthy | $2.3 \pm 13.9$ | $0.4 \pm 0.4$ | 1.1 | $100.0 \pm 0.0$ | $0.0 \pm 0.1$ | 98.9 |
| AM | MASLD | $42.7 \pm 43.2$ | $0.7 \pm 0.8$ | 88.3 | $98.7 \pm 11.3$ | $0.2 \pm 1.3$ | 11.7 |
| | MASH | $41.4 \pm 44.9$ | $0.6 \pm 1.4$ | 61.5 | $98.2 \pm 13.4$ | $0.0 \pm 0.0$ | 38.5 |
| | Healthy | $17.6 \pm 38.1$ | $0.3 \pm 2.4$ | 8.0 | $96.7 \pm 1.7$ | $0.1 \pm 0.0$ | 92.0 |
| LRI | MASLD | $76.9 \pm 41.4$ | $0.1 \pm 0.6$ | 78.6 | $92.2 \pm 3.0$ | $0.2 \pm 0.1$ | 21.4 |
| | MASH | $46.8 \pm 49.0$ | $0.5 \pm 2.0$ | 49.2 | $94.2 \pm 3.4$ | $0.1 \pm 0.1$ | 50.8 |

Table 4: F1 and AUC scores across time-gap datasets.

| Subset | Healthy | MASLD | MASH | Weighted |
|---|---|---|---|---|
| **F1** | | | | |
| Short | $0.90 \pm 0.01$ | $0.53 \pm 0.10$ | $0.45 \pm 0.05$ | $0.77 \pm 0.02$ |
| Medium | $0.90 \pm 0.00$ | $0.59 \pm 0.08$ | $0.43 \pm 0.06$ | $0.79 \pm 0.02$ |
| Long | $0.90 \pm 0.00$ | $\mathbf{0.65 \pm 0.03}$ | $0.44 \pm 0.01$ | $\mathbf{0.80 \pm 0.01}$ |
| **AUC** | | | | |
| Short | $\mathbf{0.89 \pm 0.01}$ | $\mathbf{0.95 \pm 0.01}$ | $0.78 \pm 0.01$ | $\mathbf{0.89 \pm 0.01}$ |
| Medium | $\mathbf{0.89 \pm 0.01}$ | $\mathbf{0.95 \pm 0.01}$ | $0.79 \pm 0.01$ | $\mathbf{0.89 \pm 0.01}$ |
| Long | $\mathbf{0.89 \pm 0.00}$ | $\mathbf{0.95 \pm 0.00}$ | $\mathbf{0.80 \pm 0.01}$ | $\mathbf{0.89 \pm 0.00}$ |

iron using $T_2^*$ or $cT_1$ maps, neither of which is available due to proprietary restrictions and associated costs.

**Impact of liver masking and interpretability**

Encoding the liver mask in the input has little effect on classification performance (Table 2). However, their inclusion shifts decision-making towards the liver as $P$ increases and $d$ decreases for the AM and LRI masking methods. Consequently, liver mask inclusion helps to drive disease staging independently of adjacent structures present in MRI slices, such as the kidneys, spleen, and subcutaneous fat, preventing shortcut learning and providing anatomically meaningful diagnoses. Masks also help models generalise beyond distribution shifts, including background noise and field-of-view effects, arising from differences in MRI acquisition protocols and scanners. Moreover, masking affects the network's effective receptive field, which helps to identify features from subtler texture and intensity variations within the hepatic regions consistently across images, stabilise training, and utilise shape priors (see Fig. 3).

Despite decreasing $d$ in mask inclusion, it remains non-zero, which is attributed to the expanding receptive field of stack convolutions, as features are often found near the hepatic boundary. Such identified features align with established biomarkers, which have suggested correlations between sharp hepatic boundary features and disease progression (Kim et al., 2019; Catania et al., 2021). This also highlights a growing focus (Jha et al., 2024) and an underexplored opportunity to use MRI-derived shape features to diagnose MASLD/MASH with significant fibrosis.

In the multi-modal setting, the Q-values (Table 3) reflect that the OI masking technique treats the two sequences equally across health groups, with particular emphasis on the entire liver ($T_1$) and its periphery (PDFF), where visceral fat is located. In contrast, encoding the liver mask (AM, LRI) shifts the focus to the most relevant sequence for each stage: PDFF for healthy livers, $T_1$ for MASLD, and to both for MASH with significant fibrosis. Encoding the liver mask additionally prioritises specific regions of the right (PDFF) and left ($T_1$) liver lobes, as shown in Fig. 3.

### Effect of temporal gaps

Onto the quantification of model-prediction impairment due to temporal misalignment, using Kruskal-Wallis tests (Kruskal and Wallis, 1952), followed by Dunn's post-hoc comparisons (Dunn, 1964) based on the per-health-class AUC, we find no significant differences across the short-, medium-, and long-time-gap cohorts. These results indicate that temporal gaps up to ∼12.9 years do not materially affect classification performance. This is also reflected in the comparison of aggregated metrics across health groups, with minimal differences in AUC and $F_1$ when accounting for class imbalance via weighted estimates.

### 2D vs 3D comparison

In comparisons between single-slice and volumetric inputs, our findings show that 3D models achieve performance comparable to, or slightly inferior to, their 2D counterparts in both single- and multi-modality settings (see Table 1). This reduction in performance is evident in the weighted AUCs for the fat + water input (0.83 vs 0.86) and for the PDFF input (0.84 vs 0.86).

We argue that the absence of volumetric $T_1$ images hampers 3D predictions, particularly in MASH. However, minor performance degradation still occurs in other volumetric inputs, which is attributed to the greater difficulty of learning representations in 3D deep learning, which can constrain generalisation when training data are limited. The synergy between $T_1$ and PDFF—a combination that is only possible in 2D due to data availability—is what dictates the overall superiority of the 2D approach in our study. These conclusions could change in a fully balanced comparison, which would require a dataset containing disease severity labels for MASLD/MASH, 2D, and 3D MRI images of PDFF and $T_1$; However, to the best of our knowledge, such a dataset does not currently exist.

We further suggest that discrepancies in the processing pipelines for 2D and 3D PDFF-derived from the three-point Dixon approach and the methods reported in (Langner et al., 2020), respectively, may also contribute to this gap, particularly because the latter involves intermediate steps that affect the image content, such as graph-cut–based registration of scan segments into a single contiguous volume. Although we attempted to mitigate these preprocessing differences, the performance gap persists, despite efforts to address it with either intensive data augmentation, such as intensity and contrast alterations, or with no data augmentation at all. Overall, our results across image data for UKBB indicate that single-slice PDFF and $T_1$ images with liver coverage outperform the other available sequences for diagnostic purposes.

### Key contributions and future work

Our study introduces a novel framework for generating MRI-based evidence from real-world data and highlights its underlying challenges. Our key contributions include MASLD and MASH with significant fibrosis risk stratification in connection to image-derived biomarkers, utilising deep learning explainability both quantitatively through metrics and qualitatively through visualisations of saliency maps. A combination of $T_1$ and PDFF performs best for diagnosis in UKBB, despite a high label-noise setting, demonstrating transferable learning for future MRI studies of MASLD and MASH with significant fibrosis. To encourage rapid adoption in clinical practice, our research has invested substantial effort to ensure

transparent decision-making that is easy for medical practitioners to understand and free of systematic bias. The identified consistent activations in specific liver regions motivate our future research to focus on identifying shape biomarkers that link liver morphology to MASLD/MASH diagnosis.

However, our approach applies to a wider range of clinical uses beyond MRI and liver disease. Components of our analysis could be utilised towards the analysis of deep learning systems' quality assurance, where statistical analysis of saliency maps can confirm that current clinical algorithms are consistent with pathophysiology. Additionally, understudied medical data could benefit from our methodological application, especially in settings where significant label noise prohibits alternative approaches and allows only dataset-level explainable statistical aggregation. While several methodology components are transferable, implementations would require modifications in explainability strategies, handling of label noise, and disease labelling, based on data availability.

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

## Appendix A. Liver Localisation in volumetric images

The liver occupies only a small fraction of both single-slice and volumetric UKBB MRI images. In light of this, providing liver regions is expected to help the deep learning pipeline focus its decision-making on relevant areas while significantly reducing computational and memory costs. In single-slice images ($T_1$, PDFF), we use segmentation to encode liver location. In contrast, for volumetric images (fat + water, PDFF), we prefer three-dimensional

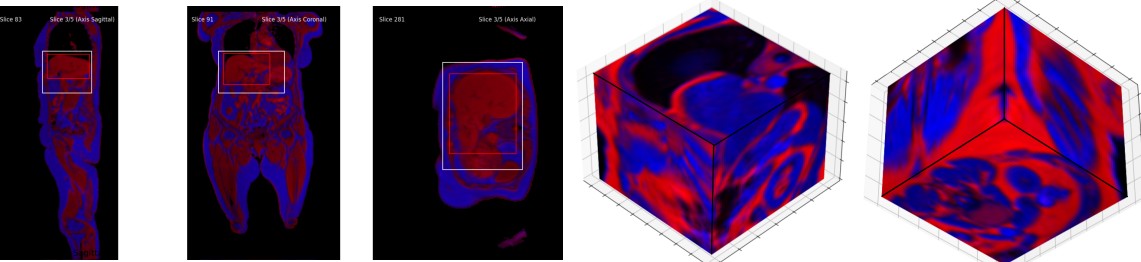

Figure 4: A preview of a single slice per axis of the developed GUI for generating liver bounding box annotations (left figures). The user can add explicit liver location annotations (red rectangles) based on the approximate liver bounding box predictions (white rectangles). Upon completion of annotations, the user is presented with a visualisation of the proposed BB's boundaries (right figures) and can confirm or adjust the annotations.

bounding box detection because their annotation requires less effort. 2D annotations are obtained from Al-Belmpeisi et al. (Al-Belmpeisi et al., 2024), and in this current work, we focus on 3D bounding box detection models.

We start by manually annotating the liver bounding box on 100 volumetric images. These annotations are used to train the approximate liver detection network using an $80/10/10$ train/test/validation split. The approximate liver detection network accelerates annotation via a graphical user interface (GUI) we have developed, a snapshot of which is shown in Fig. 4. The GUI displays an approximation of the liver bounding box across multiple slices in all axes, which the user can refine by adding annotations. After the refined rectangles are added, the smallest parallelepiped that fully contains all annotations is computed and used as a refined liver bounding box annotation.

After manual annotation, we use the GUI to annotate $N = 20$ slices per axis, yielding 500 annotated liver volumes. These annotations are used to train the refined liver detection network, with a $400/80/20$ split for the training, validation, and test sets, respectively.

All liver location networks are trained on volumetric fat and water channels to predict the bounding box limits ($x_c$, $y_c$, $z_c$, $\Delta x$, $\Delta y$, $\Delta z$), based on a combination of the Complete Intersection over Union (CIoU) loss enhanced by two additional novel loss components, the liver loss and boundary loss. The final loss reads as follows:

$$L = L_{\text{CIoU}} + L_l + L_b. \tag{2}$$

CIoU loss is of utmost importance, as it optimises Intersection over Union (IoU), central point distance, and aspect ratios (Zheng et al., 2020). This ensures meaningful weight updates, even when there is no overlap between the prediction and ground truth, which can occur during the initial epochs and destabilise training.

The liver loss $L_l$ contribution is expressed as:

$$L_l = \sum_{p \in \{x,y,z\}} \left[ \text{ReLU} \left( p_{\min} - p_{\min}^{GT} \right) + \text{ReLU} \left( p_{\max}^{GT} - p_{\max} \right) \right], \tag{3}$$

where $p$ and $p^{GT}$ refer to the coordinates of the predicted and ground truth box, respectively, and for the Rectified Linear Unit (ReLU), we have $ReLU(x) = \max(0, x)$. The objective of liver loss is to maximise detection recall, i.e., to increase the likelihood that the complete liver is fully contained within the predicted box; thus, it is zero only when the liver is fully contained within the predicted box.

The equation for the boundary loss $L_b$ reads as:

$$L_b = \sum_{p \in \{x,y,z\}} \left[ \text{ReLU}(-p_{\min}) + \text{ReLU}(1 - p_{\max}) \right]. \qquad (4)$$

The boundary loss $L_b$ penalises the predicted bounding box whenever it exceeds the volumetric image limits $(0, 1)$, which can occasionally occur, for example, if we acquire large $x_c$ and $\Delta x$ predictions for a volume.

All liver detection networks are trained with a learning rate scheduler that reduces the learning rate by 50% every 5 epochs, without increasing validation CIoU, and early-terminates after 10 epochs without an increase in validation CIoU. After training for 15 epochs, the approximate liver detection network achieves a test IoU of 0.53, which is sufficient for the GUI implementation and for helping in providing additional annotations. Provided by the additional annotations, the networks achieve a mean test IoU of $0.80 \pm 0.12$, with a predicted box centre error $(\Delta d_c)$ of $11.24 \pm 12.24$ mm, which translates to $\Delta x_c = -1.58 \pm 5.62$ mm, $\Delta y_c = 2.85 \pm 12.43$ mm, and $\Delta z_c = -1.66 \pm 9.40$ mm for the coronal, sagittal and axial axes, respectively.

Our preprocessing pipeline employs padding to include a broader volumetric region around the detected box to mitigate the impact of errors in our liver bounding box detection models. To guarantee that the liver is always completely included in the volumetric inputs, the model's volumetric inputs are enlarged by 10% across all dimensions. Liver detection effectively reduces volumetric image size by about 67%, which is crucial for overcoming GPU memory issues and the computational limitations of the pipeline's following classification step.

## Appendix B. Implementation details

We use an 18-layer ResNet implementation from https://github.com/xmuyzz/3D-CNN-PyTorch, to process both single-slice and volumetric images. For volumetric images, all convolutional and pooling operations are replaced with their volumetric counterparts, which enable feature extraction from the entire 3D image volume. The architecture consists of four convolutional stages; Each stage is composed of two basic residual blocks, followed by identity skip connections. The channel dimensions of each convolutional stage progressively increase (64, 128, 256, 512), while downsampling is performed at the first block of each stage. The final convolutional stage uses adaptive average pooling, followed by a fully connected layer, which maps the 512-dimensional convolution-based features to the output classification layer of three classes. All layer weights were initialised using Kaiming normal initialisation (He et al., 2015).

We use the Adam optimiser (Kingma, 2014) with a learning rate of $10^{-3}$ and a batch size of 4 for both volumetric and single-slice inputs in our studies. Compared to the 2D

pipeline ($\approx$11 million parameters), the higher kernel dimensionality of 3D convolutions results in models with a larger feature extraction capacity for volumetric images ($\approx$33 million parameters). Contrary to one hour for the 2D pipeline, training on 3D images takes about six hours, on an NVIDIA L4 GPU with 24 GB of VRAM.

