# OpenReview forum: "Deep Learning for Liver Disease Stratification: Findings from UKBB MRI"
_MIDL.io/2026/Validation_Papers — MIDL 2026 - Validation Papers Poster_

### Official Review · Reviewer_Mfcv · 2026-01-04

**Confidence:** 3
**Preliminary Rating:** 3
**Final Rating:** 3

**Summary:**

This work investigates deep learning diagnosis of MASLD and MASH with significant fibrosis using MRI data from the UK Biobank. The authors compare 2D versus 3D inputs, single-modal versus multi-modal MRI sequences, and different inputs (full image, mask-augmented, and liver-isolated). A key idea is avoiding shortcut learning by evaluating how liver segmentation masks influence model attention. Experiments show that: 1) 2D inputs achieve comparable or better performance than 3D. 2) While liver masking has minimal impact on AUC, it redirects model attention toward relevant regions.

**Strengths:**

1. The experimental design is well-motivated. The comparison across 2D/3D, single-/multi-modal inputs, and masking strategies is thorough and directly addresses clinical  trade-offs between performance, interpretability, and cost.
2. The authors pinpoint the key issue in deep learning based diagnosis: explainability and shortcut, by quantitatively and qualitatively assessing whether models focus on liver regions, which is critical for clinical trust.
3. The paper provides clear practical insights. The conclusion that 2D MRI is sufficient and often preferable provides useful insights for future model design and clinical deployment.

**Weaknesses:**

1. The models explored in the paper are quite limited. With only 18-layer ResNet, it is unclear if the conclusion is widely applicable. It would be better to explore more recent architectures or attention-based models commonly used in medical imaging.
2. The explainability method is also limited. The paper primarily relies on back propagation based explainability model, which may not fully capture the model behavior, given known limitations of gradient-based saliency methods.
3. Although the authors use some augmentations, the 3D setup is constrained by voxel anisotropy and limited modality choices, which may underrepresent the potential of volumetric modeling. This may explain why the 3D models does not bring about additional benefits compared to 2D methods.

**Detailed Comments:**

1. Thresholds such as the top 1% activation criterion and specific augmentation probabilities could be better motivated or supported by ablation studies.

**Justification Of Final Rating:**

Thanks for the responses. The rebuttal makes some explanation on the key concerns, which addresses some of my concerns. The study provides both quantitative results and explainability analyses, highlighting practical considerations for real-world MRI-based liver disease assessment. Overall, I do not object to the acceptance.

**Justification Of The Preliminary Rating:**

The study provides both quantitative results and explainability analyses, highlighting practical considerations for real-world MRI-based liver disease assessment. Overall, the manuscript is well prepared, but the models, explainability tools, and configurations used in the paper are limited.

**Questions To Address In The Rebuttal:**

The study acknowledges label noise due to the temporal gap between MRI acquisition and biomarkers used for MASLD/MASH stratification. Can the authors quantify how performance varies as a function of the time gap, and clarify to what extent the noise issue is mitigated?

---

### Official Review · Reviewer_Epnk · 2026-01-09

**Confidence:** 4
**Preliminary Rating:** 4
**Final Rating:** 5

**Summary:**

The authors present a deep learning framework for liver disease stratification in metabolic dysfunction-associated steatotic liver disease (MASLD) and metabolic dysfunction-associated steatohepatitis (MASH). Using multi-modal MRI inputs (fat, water, T1, and PDFF maps), they train a ResNet-18 classifier and compare 2D versus 3D imaging configurations for disease classification. The study leverages a large cohort of 18,073 subjects (11,706 healthy; 4,254 MASLD; 2,113 MASH with significant fibrosis). The results suggest that 2D MRI provides sufficient diagnostic performance, while 3D offers no clear advantage at higher computational cost. Overall, the work offers a large-scale validation of an MRI-based pipeline for liver disease classification with potential translational value for related clinical decision-support applications.

**Strengths:**

- **Translational Relevance**: The paper introduces a deep-learning framework for liver disease stratification (MASLD/MASH) that could be transferable to related MRI-based diagnostic workflows in clinical practice.
- **Comprehensive Evaluation**: The experimental validation is extensive, leveraging a large cohort across three clinically meaningful groups (11,706 healthy; 4,254 MASLD; 2,113 MASH with significant fibrosis) and comparing both 2D vs 3D imaging configurations using multiple MRI-derived images (fat, water, T1, PDFF).
- **Study Design**: The interpretability analysis is a useful addition where the defined saliency map metrics (P, Q, and d) help quantify and compare the effect of masking beyond overall classification accuracy.
- **Clinical Relevance**: The clinical motivation and target application (e.g., MASH/MASLD) are clearly articulated, making the problem setting and potential impact easy to follow.
- **Presentation**: The manuscript is well organized and visually clear, with strong supporting figures/tables (pipeline overview, cohort characterization, and liver segmentation/visual examples) that make the experimental setup and results easy to interpret.

**Weaknesses:**

- Key training and compute details are missing. Please report model capacity (number of parameters), batch size for training, optimizer, training time, and GPU memory usage for both the 2D and 3D settings, along with the hardware/software stack (e.g., GPU model).
- The conclusion that “3D provides no benefit over 2D” would be stronger if stated more cautiously and supported with clearer justification. Because 2D vs 3D comparisons can be confounded by differences in acquisition/protocol, preprocessing/labeling (e.g., annotations vs bounding boxes), and using the same backbone architecture (ResNet-18) for both settings, it is difficult to interpret performance differences as purely attributable to dimensionality (2D vs 3D). It would help to (i) describe more clearly what is held constant vs different between 2D and 3D experiments, and (ii) cite prior studies that report similar findings in these situations.
- While masking does not appear to improve overall classification accuracy, it does shift model attention (as reflected by the saliency metrics in Table 3 and Fig. 3). It would strengthen the paper to clarify what practical benefit this provides, for example, whether masking can reduce training cost (fewer epochs, faster convergence) or improve data efficiency (achieving similar performance with less training data). Otherwise, given sufficient data/epochs, a ResNet-18 may learn similar cues from the original unmasked data alone, which makes the added value of masking less clear. An ablation study varying the number of training epochs and/or training-set size would help support this claim. This is especially useful for computationally expensive 3D training.

**Detailed Comments:**

- In Table 1, please also highlight the MASH results for the T1-only setting (first three rows), not only the T1+PDFF configuration.
- If you used complex-valued T1 MRI images, please clarify the representation used in the network (e.g., real/imaginary as two channels).
- It would be helpful to comment on other potential applications of this framework beyond liver stratification and beyond MRI, and what adaptations would be needed for those settings.

**Justification Of Final Rating:**

The authors addressed my main concerns in the rebuttal. They improved reproducibility by adding missing training/computational details, refined the 2D vs. 3D discussion by explicitly acknowledging confounders and softening the conclusion to a practical statement tied to dataset constraints, and clarified the motivation for liver masking. Overall, the work is clinically relevant, well validated at scale, and I recommend acceptance.

**Justification Of The Preliminary Rating:**

The paper has sound validation-related strengths such as clear clinical and translational relevance, comprehensive evaluation and great presentation/demonstration. However, it lacks computational details and explanation/justification of conclusions.

**Questions To Address In The Rebuttal:**

Address weaknesses:
- Please report model capacity (number of parameters), batch size for training, optimizer, training time, and GPU memory usage for both the 2D and 3D settings, along with the hardware/software stack (e.g., GPU model).
- The conclusion that “3D provides no benefit over 2D” would be stronger if (i) describe more clearly what is held constant vs different between 2D and 3D experiments, and (ii) cite prior studies that report similar findings in these situations.
- Explain advantage of masking in detail.

---

### Official Review · Reviewer_bsbi · 2026-01-10

**Confidence:** 5
**Preliminary Rating:** 4
**Final Rating:** 5

**Summary:**

The author compared performance of 2D and 3D 18 layers resnet on diagnosing MASLD and MASH with significant fibrosis using MRI sequences (specifically Fat+Water and PDFF) from the UK Biobank dataset. Shows that 3D models achieved performance comparable to or slightly inferior to their 2D counterparts, leading to the conclusion that single-slice 2D inputs are sufficient and more computationally efficient for this task.

**Strengths:**

The authors invested significant effort in designing the experiments and model training to ensure a fair comparison between 2D and 3D ResNets. Strategies to prevent overfitting, such as early stopping, were also implemented, making the results more reliable.

**Weaknesses:**

The claim that the 3D model is worse than the 2D model may be caused by differences in how the images were prepared and processed and how the input of the model is designed, rather than just the use of 3D data itself. Specifically, the 3D images went through intermidiate steps and these steps may have altered the image details as the author metioned in the paper. Additionally, the 3D method relied on an automatic tool to find and crop the liver that was not perfect (about 80% accuracy), which likely introduced errors that the 2D method avoided.

**Detailed Comments:**

1. You may want to define the meaning of 3D masking in the table 1.
2.To enhance reproducibility, please provide the code for the paper or specify the details of the network structures in the Appendix. A reference to an '18-layer ResNet architecture' is too general.

**Justification Of Final Rating:**

The authors’ rebuttal and revisions addressed my concerns. The 2D vs 3D preprocessing and bounding-box handling are now clearly explained, table 1 and masking terminology are clearer, and the added network details make the work easier to reproduce.

**Justification Of The Preliminary Rating:**

This paper uses a large dataset to show that simple 2D images are effective for diagnosis, which is a useful and practical finding. However, the comparison is not completely fair because the 3D images were processed differently than the 2D ones and the design of model inputs may effect the networks' performance (see line 6, 9 and 12 in table 1), making the conclusions about 3D performance less reliable.

**Questions To Address In The Rebuttal:**

In table 1, for line 6, 9 and 12:
Data Masking Healthy MASLD MASH
PDFF LRI 0.86 ± 0.00 0.94 ± 0.01 0.72 ± 0.01
T1 + PDFF LRI 0.88 ± 0.01 0.95 ± 0.01 0.77 ± 0.02 0.88 ± 0.01
PDFF 3D 0.85 ± 0.01 0.89 ± 0.01 0.73 ± 0.01

Is it possible that the performance advantage of the 2D model is driven by the availability of $T_1$ inputs

---

### Author Rebuttal · Authors · 2026-01-24

**Rebuttal:**

In response to the reviewers’ feedback, we have updated the manuscript and appendices to improve transparency and reproducibility, and discussed potential misunderstandings of our methodology.


On Reviewer 1's comments:

Preprocessing: To ensure full liver recall, Appendix A has been updated to more precisely describe the padding strategy used to address bounding box detection errors (IoU ~80%).

Terminology: To distinguish between mask-based and segmentation-based techniques, Table 1 was updated from "3D" to "BB" (Bounding Box).

Reproducibility: Appendix B was added, containing hyperparameters, specifics of the 18-layer ResNet architecture, and a link to the backbone's GitHub repository.

Results formatting: To consistently highlight the best models across health classes, including T1-MASH, Table 1's bolding was adjusted.

Discussion: Acknowledged that the unavailability of T1 3D sequences in UKBB contributes to the 2D performance advantage.

On Reviewer 2's comments:

Compute Details: NVIDIA L4 hardware, model capacity (11M vs. 33M parameters), optimiser settings, and training times were added to Appendix B.

Refined Conclusions: The discussion was updated to make it easier to follow commonalities and differences between 2D vs 3D (preprocessing/T1 availability). Softened conclusions to highlight 2D as a practical option rather than a theoretical benefit, given data availability.

Interpretability: Extended the discussion to clarify that the advantage of liver masking is anatomical validity and "shortcut learning" mitigation rather than performance gains.

Future Work: We added a paragraph about potential translational applications of our framework's components, including other pathologies with noisy labels, or quality assurance of deployed AI models in the clinic.

On reviewer 3's comments:

Anisotropy: Explained how resampling is the only strategy allowing 3D rotations, which is prohibitive in the anisotropy of MRI data. In essence, our strategy truly is in line with most reliable methods for augmenting MRI data without introducing artefacts, while also respecting the underlying data.

Saliency: Explained that cohort-level saliency aggregation acts as a natural denoising mechanism for attributions, and Fig. 2 eliminates vanishing gradient concerns, suggesting that gradient-based XAI is reliable.

Label Noise: Referenced Table 4 and Section 3.2, for a quantitative analysis (H-test) demonstrating model stability over temporal gaps of up to 12.9 years.

**Supporting Material:**

/attachment/8838c76520a2cfd5776d45c3973781f0e3ce195f.pdf

---

### Meta-Review · Area_Chair_pnMa · 2026-02-04

**Recommendation:** Accept (Poster)
**Confidence:** 4

**Metareview:**

The reviewers agree the paper is clinically relevant, well presented, and supported by a comprehensive evaluation with quantitative and explainability analyses, yielding a practical finding that 2D inputs can be effective for diagnosis. While initial concerns were raised about missing computational details, limited model/configuration breadth, and the fairness of the 2D vs 3D comparison, the rebuttal substantially clarified preprocessing, bounding-box handling, and network details, improving reproducibility and strengthening the main conclusions.

---

### Decision · Program_Chairs · 2026-02-14

Accept (Poster)